# Applied Fence-Post Techniques Using Deep Electrodes Instead of Catheters for Resection of Glioma Complicated with Frequent Epileptic Seizures: A Case Report

**DOI:** 10.3390/brainsci13030482

**Published:** 2023-03-13

**Authors:** Shunsuke Nakae, Masanobu Kumon, Takao Teranishi, Shigeo Ohba, Yuichi Hirose

**Affiliations:** Department of Neurosurgery, Fujita Health University, Toyoake 470-1192, Japan

**Keywords:** deep electrodes, fence-post, glioma, frequent seizures

## Abstract

Fence-post catheter techniques are used to use tumor margins when resecting gliomas. In the present study, deep electrodes instead of catheters were used as fence-posts. The case of a 25-year-old female patient whose magnetic resonance images (MRI) revealed a tumor in the left cingulate gyrus is presented in this study. She underwent daily seizures without loss of consciousness under the administration of anti-seizure medications. Despite video electroencephalography (EEG) monitoring, the scalp inter-ictal EEG did not show obvious epileptiform discharges. We were consequently uncertain whether such frequent seizures were epileptic seizures or not. As a result, deep electrodes were used as fence-posts: three deep electrodes were inserted into the tumor’s anterior, lateral, and posterior margins using a navigation-guided method. The highest epileptic discharge was detected from the anterior deep electrode. As a result, ahead of the tumor was extendedly resected, and epileptic discharges were eliminated using EEG. The postoperative MRI revealed that the tumor was resected. The patient has never experienced seizures after the surgery. In conclusion, when supratentorial gliomas complicated by frequent seizures are resected, intraoperative EEG monitoring using deep electrodes as fence-posts is useful for estimating epileptogenic areas.

## 1. Introduction

Seizures are frequently complicated by low-grade gliomas, such as adult diffuse gliomas harboring mutant *isocitrate dehydrogenase* (*IDH*), some pediatric low-grade gliomas, or some glioneuronal and neuronal tumors [1,2,3,4]. Adult diffuse glioma patients with *IDH* mutations are younger than patients with adult diffuse gliomas without mutant *IDH* [5,6]. Patients with these low-grade gliomas sometimes experience frequent or drug-resistant seizures, which can impair decreased quality of life or even brain functions. As a result, in such cases, glioma surgery should focus not only on reducing mass as much as possible while preserving brain functions but also on improving seizure prognosis to lead a fulfilling social life. Several methods have been developed to achieve the safest resection of gliomas, including awake craniotomy, the use of 5-aminolevulinic acid, and the navigation-guided fence-post method [7,8,9,10,11]. The navigation-guided fence-post technique can be used to create a visible margin of glioma before brain shift occurs during surgery. Previous studies reported its effectiveness, especially in improving postoperative functional outcomes in patients with glioblastoma [11,12]. On the other hand, intraoperative electrocorticography (ECoG) is commonly used in epilepsy surgery to improve postoperative seizure prognosis. The utility of ECoG for drug-resistant epilepsy associated with supratentorial lesions such as cavernomas and gliomas in improving seizure prognosis or detecting epileptogenic foci has also been reported [13,14].

In this study, the navigation-guided fence-post technique was combined with continuous EEG monitoring during surgery for a patient with low-grade diffuse glioma complicated by frequent seizures, using deep electrodes as fence-posts instead of catheter tubes for epileptic seizure diagnosis and the estimation of epileptogenic areas. This paper describes how this novel method can be used to achieve maximum safe resection while also improving seizure prognosis in tumors complicated by frequent or drug-resistant seizures.

## 2. Case Presentation

A 25-year-old right-handed female patient visited a hospital due to frequent seizures that started a couple of months ago. She was consequently referred to our institution after magnetic resonance imaging (MRI) revealed a mass suspected of being a glioma in the left cingulate gyrus (Figure 1A). When she visited our institution for the first time, she underwent focal onset aware seizures more than 10 times during the medical interview. The symptom was focal face clonic movement and head turning to the right side for 20–30 s. According to the patient, the seizures were frequent during her menstrual cycles. Although she has never been diagnosed with developmental delay or disorders, her Wechsler Adult Intelligence Scale (WAIS) and Wechsler Memory Scale-revised (WMS-R) scores were significantly lowered. WAIS scores were: 57 for full-scale intelligence quotient, 70 for verbal comprehension index, 56 for perceptual organization index, 74 for working memory index, and 66 for processing speed index. The scores of WMS-R were as follows: verbal memory 52, visual memory 55, generalized memory <50, attention concentration 60, and delayed recall <50. Even though electroencephalography (EEG) revealed no obvious epileptic discharges, an anti-seizure medication (ASM) was administrated. After the administration of ASMs was gradually increased, seizure frequency was improved. Despite taking 2000 mg of levetiracetam and 4 mg of perampanel, she had seizures more than ten times per day. We then conducted continuous video EEG monitoring before tumor resection. The seizure symptom was frequently observed, but the seizure onset zone was unclear according to the scalp EEG, presumably due to the thickness of the frontal bone (Figure 1B). At this time, we were uncertain whether her seizure symptom was an epileptic seizure or a non-epileptic seizure, such as a psychogenic non-epileptic seizure. As a result, instead of catheters, we used deep electrodes as fence-posts to confirm whether her symptom was caused by epilepsy associated with glioma and to detect the epileptogenic zone during surgery.

The following are the surgical procedures: During the surgery, the EEG (EEG-1260; Nihon Kohden, Co., Tokyo, Japan), motor-evoked potentials (MEP), and somatosensory-evoked potentials (SEP) were used for neuromonitoring. The patient was placed in the supine position. The head was fixed in the median and vertex-up positions after general anesthesia was induced. Sevoflurane was used to maintain anesthesia so that epileptic discharges could be easily detected with EEG. The reference electrode was placed on her forehead for the referential recording of EEG. The brain surface was exposed following the frontal craniotomy. First, the location of the central sulcus was identified using MEP and SEP to not injure the precentral sulcus. Then, using the navigation-guided method, three deep electrodes with six electrodes each (Unique Medical Co., Ltd., Tokyo, Japan) were inserted into the tumor’s anterior, posterior, and lateral margins (Figure 2). The depth of each deep electrode corresponded to the depth of the tumor: the depths of anterior, posterior, and lateral electrodes were 39, 40, and 35 mm from the brain surface, respectively. We used the following montages: referential recording, bipolar recording, and average potential reference. As a result, numerous epileptic discharges were detected according to referential recording and average potential reference (Figure 3A), and the highest peak originated from the fourth electrode of the anterior deep electrode according to bipolar recordings (Figure 3B,C). The tumor was resected using the superior frontal gyrus as a surgical window. Based on the EEG results, ahead of the tumor was extendedly resected because the anterior cingulate gyrus and superior frontal gyrus are considered non-eloquent areas. Epileptic discharges were eliminated following tumor resection using a subdural grid electrode placed on the brain surface around the cavity (Figure 4). The MEP did not reveal any changes during the surgery.

The patient showed right hemiparesis and aphasia after the surgery. A postoperative MRI revealed that the tumor had been completely resected (Figure 5A) and that there were no cerebral infarctions that could have affected her motor and speech functions. These symptoms were improved two weeks after the surgery and were consequently considered to be due to the supplementary motor area syndrome. The tumor’s histology was diffuse astrocytoma and the histological image of the hematoxylin and eosin stain was shown in Figure 5B. *IDH1* mutations were not found using immune histological staining or the Sanger’s method. The patient was discharged without any neurological deficits nineteen days after the surgery. Copy numbers were analyzed by next-generation sequencing (NGS). The methods of the copy number analysis were described in our previous study [15]. Briefly, the extracted DNA was amplified by whole genome amplification SurePlex. Each sample was tagged and fragmented, and then prepared for NGS analysis. After purification and normalization of the DNA libraries, the libraries were sequenced using VeriSeq PGS Kit-MiSeq. The copy number analysis revealed that chromosome 6q had been lost, but chromosomes 1p and 19q remained intact (Figure 5C). These molecular examinations suggested that the tumor might be classified as pediatric-type diffuse low-grade glioma, diffuse astrocytoma, *MYB*- or *MYBL1*-altered, which frequently causes drug-resistant epilepsy [4]. The tumor has not recurred after the surgery without any postoperative therapy for more than one and a half years and she has never experienced seizures after the surgery.

## 3. Discussion

In this study, deep electrodes were used as fence-posts to resect a diffuse glioma complicated by frequent focal awareness seizures; gross total resection of the tumor as well as being long-term seizure-free was achieved for the patient. This technique provides us with information about not only a visible border of diffuse glioma before a brain shift occurs but also continuous EEG patterns. To the best of our knowledge, this is the first report to use deep electrodes as fence-posts instead of catheters to resect a tumor that causes frequent seizures.

It is well known that gliomas, especially low-grade gliomas, are frequently complicated with epileptic seizures. Previous research has found that the semiology of seizures, particularly those from the frontal lobe or cingulate gyrus, is difficult to characterize due to their diversity [16,17], making diagnosis difficult at times. In this case, the patient’s seizure symptoms included focal face clonic movement bilaterally, slightly turning her face to the right side, and occasionally laughing. She had never experienced unawareness seizures although she underwent highly frequent seizures. Furthermore, the scalp EEG revealed no obvious epileptic discharges. However, her scores of WAIS and WMS-R were extremely decreased despite normal development in her childhood. Taking these factors into account, we concluded that her symptom was most likely caused by tumor-related epilepsy, but non-epileptic seizures were also possible. If the symptom was a truly epileptic seizure, such highly frequent seizures should have been stopped by the surgery. As a result, we attempted to use deep electrodes as fence-posts for diagnosing and detecting the loci of the most irritative area that was thought to be associated with the epileptogenic area. Accordingly, the intraoperative EEG indicated frequent epileptic discharges, and the fourth electrode of the anterior deep electrode (A4 in Figure 3) was the highest peak. We extended the resection around the electrode, and the patient has never had seizures since the surgery. The intraoperative EEG revealed that the epileptogenic focus is the anterior cingulate gyrus. According to a report on the semiology of cingulate gyrus epilepsy, hyper motor seizures are frequently observed in patients with anterior cingulate gyrus epilepsy [17]. In this case, her symptom was thought to be classified as a focal awareness seizure with face clonic movement and head turning, which might be similar with hyper motor seizures.

The navigation-guided fence-post technique is used in glioma surgery as a guide for tumor margin before brain shift. Catheter tubes are generally used for fence-posts, but we used deep electrodes in the present case. Only one previous study used electrodes as fence-posts instead of catheters, but electrodes were used for functional mapping during surgery in this study [18]. To the best of our knowledge, this is the only report that deep electrodes were used as fence-posts for detecting the most irritative zone during surgery.

The current study has some limitations. First, this is the only case that shows the long-term effects of the use of deep electrodes as fence-posts for both the total resections of tumors and the identification of irritative areas. This fence-post technique has been used for the resection of supratentorial gliomas or lesions that have been complicated with frequent or drug-resistant epilepsy since this case. The usefulness of this technique will be examined for our case series as a future study. Second, deep electrodes used during surgery showed interictal patterns, so areas around the electrode with the highest peak were thought to be an irritative zone that could be associated with an epileptogenic zone. In the current study, the tumor was removed along this line and it was confirmed that the epileptic discharges had stopped using a subdural electrode. It may be more effective if the disappearance of epileptic discharges is confirmed while still using deep electrodes because the confirmation of disappearance is more reliable when performed using the same electrode. Therefore, this method still has room for modification for the estimation of epileptogenic areas during surgeries for future studies.

## 4. Conclusions

Intraoperative EEG monitoring with deep electrodes as fence-posts is beneficial not only for tumor resection but also for estimating epileptogenic areas for the resection of a supratentorial lesion complicated by frequent or drug-resistant epilepsy. However, this method should be modified for the certain identification of the epileptogenic loci during the surgery.

## Figures and Tables

**Figure 1 brainsci-13-00482-f001:**
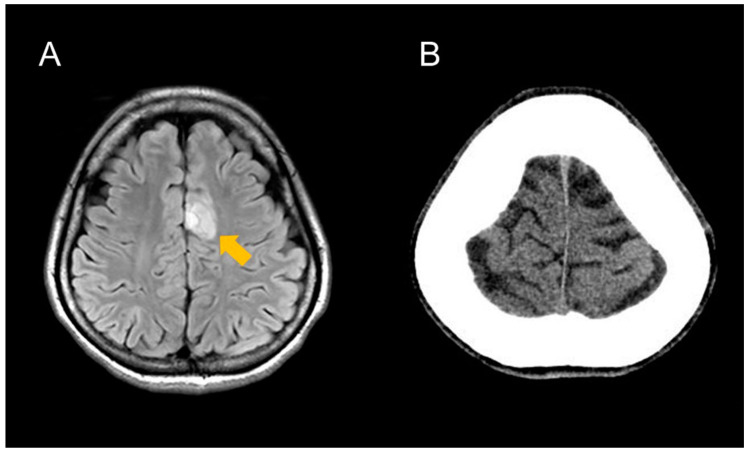
Preoperative MRI (**A**) and CT (**B**) images of the patient. All magnetic resonance examinations in this study were performed with a 3-T MR system (Vantage Centurian, Canon Medical Systems Corporation, Otawara, Japan). The fluid-attenuated inversion recovery (FLAIR) image revealed high intensity mass (arrow) in the left cingulate gyrus. The CT image revealed that the maximum thickness of the frontal bone was 28 mm, which might be associated with the epileptic discharges being unclear based on scalp EEG.

**Figure 2 brainsci-13-00482-f002:**
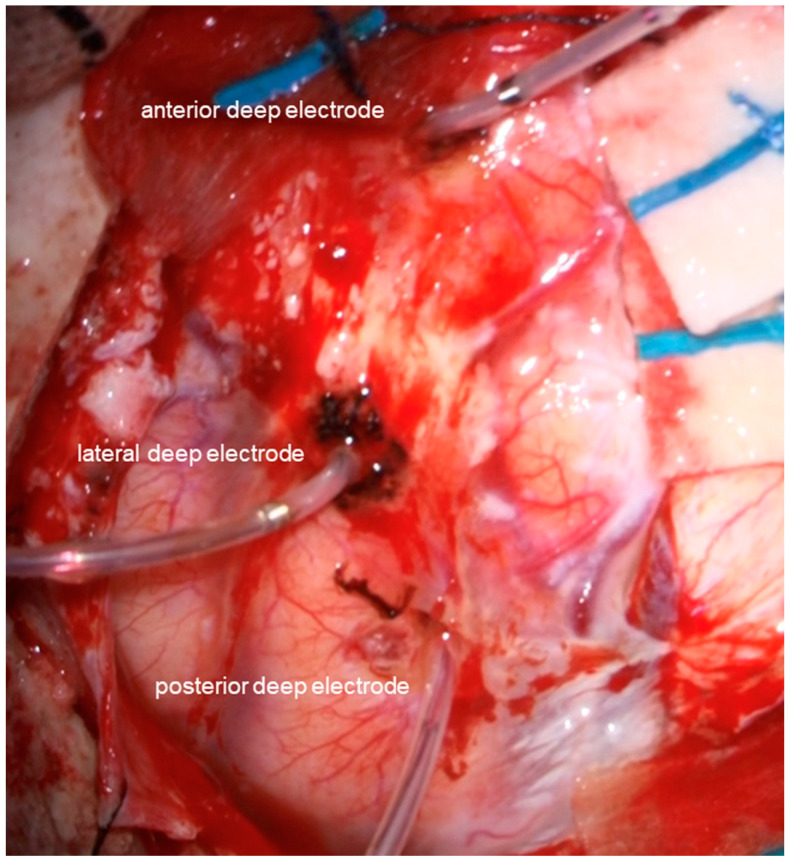
An image taken intraoperatively following the insertion of deep electrodes. Using a navigation-guided technique, three deep electrodes were placed along the tumor’s anterior, lateral, and posterior edges.

**Figure 3 brainsci-13-00482-f003:**
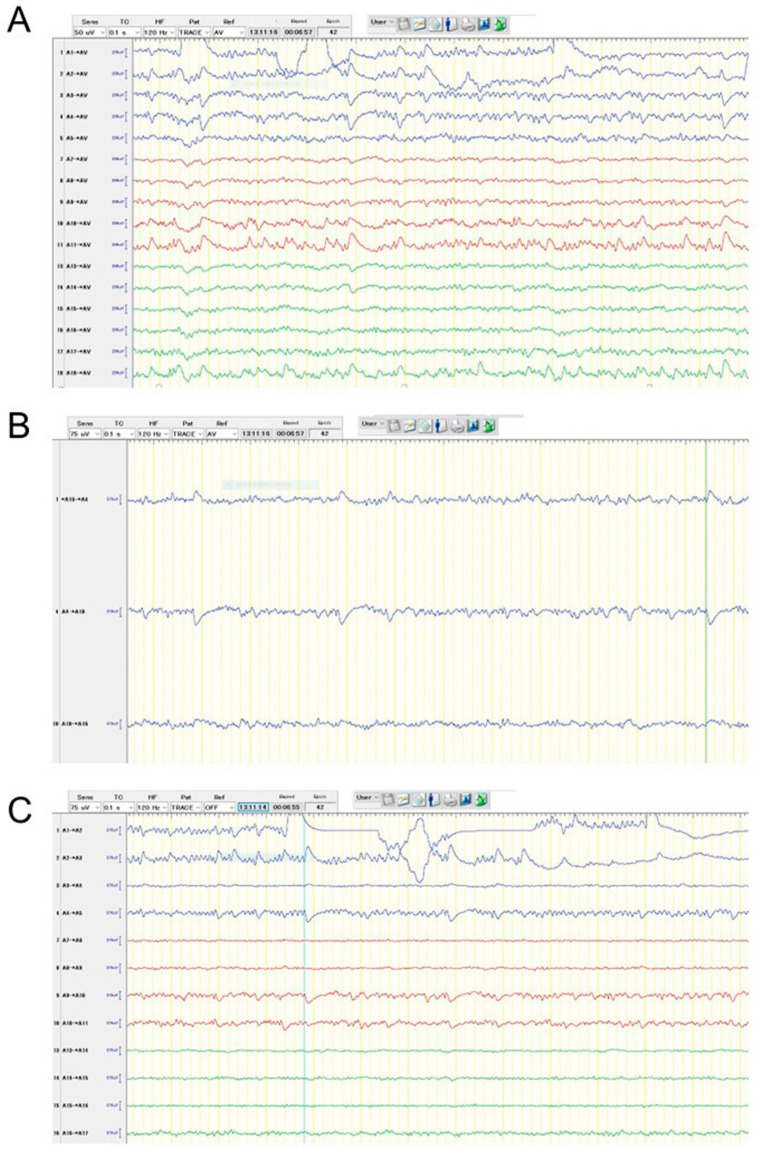
Intraoperative EEG patterns before resection of the tumor according to the different montages: average potential reference (**A**), bipolar recording in similar levels of depth among three deep electrodes (**B**), and bipolar recording in each deep electrode (**C**). There were 6 electrodes in each deep electrode: A1–A6 were from the anterior deep electrode (blue), A7–A12 were from the lateral deep electrode (red), and A13–A18 were from the posterior deep electrode (green). Of these electrodes, A6 and A12 which included much noise were excluded from the figures. According to the average potential reference pattern, the fourth electrode of the anterior deep electrode seemed to have the highest peak (**A**). We evaluated three deep electrodes using bipolar recording to demonstrate that the anterior deep electrode displayed the biggest peak (**B**). Accordingly, the phase reversal was identified between A16–A4 and A4–A10, showing that A4 from the anterior deep electrode was the highest peak. Finally, bipolar recording inside the anterior deep electrode showed that phase reversal was identified between A3–A4 and A4–A5. Taken together, the area around electrode A4 was considered to be the irritative zone.

**Figure 4 brainsci-13-00482-f004:**
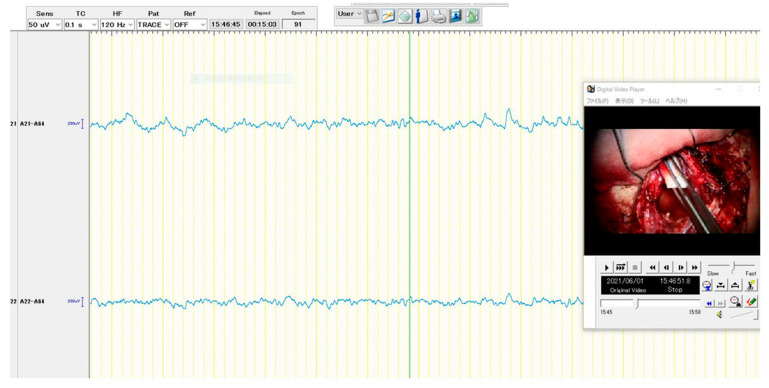
Intraoperative EEG patterns after resection of the tumor according to the referential recording. As a picture inside the figure depicts, a subdural grid was placed on the surface of the brain ahead of the resected cavity. The apparent epileptic discharges were not detected from this record.

**Figure 5 brainsci-13-00482-f005:**
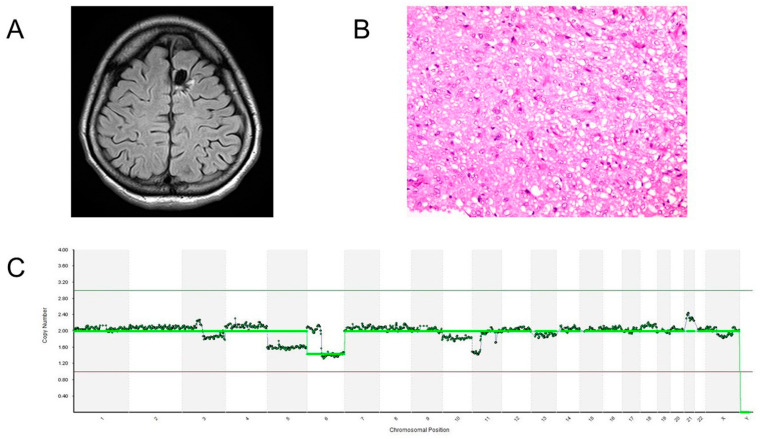
Postoperative radiological, histological, and molecular examinations. (**A**) A postoperative FLAIR image reveals that the tumor was totally resected. (**B**) The histological image of hematoxylin-eosin stain for the resected tumor is demonstrated. (**C**) The figure indicates the result of copy number analyses based on NGS for the resected tissue. A vertical axis indicates the calculated copy number and a horizontal axis shows the chromosomal number.

## Data Availability

Not applicable to this study.

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
