# Peer review of "Applied Fence-Post Techniques Using Deep Electrodes Instead of Catheters for Resection of Glioma Complicated with Frequent Epileptic Seizures: A Case Report"

_brainsci, 2023, doi:10.3390/brainsci13030482_

Round 1
Reviewer 1 Report
Mnuscript entitled "Applied fence-post techniques using deep electrodes instead of catheters for resection of glioma complicated with frequent epileptic seizures: A case report"
This work is of interesting while some issued should be addressed to make it more attractive:
1. The representative HE of the tumor and the margin of excision should be disclosed.
2. The methods (assay, panel) used for NGS test should be mentioned in more detail. And the findings of NGS should be summarized.
3. The NGS test should include FGFR fusion and NTRK fusion.
Author Response
The reviewers’ comments have helped us improve our manuscript. We have attempted to address all questions raised by the reviewer.
Mnuscript entitled "Applied fence-post techniques using deep electrodes instead of catheters for resection of glioma complicated with frequent epileptic seizures: A case report"
This work is of interesting while some issued should be addressed to make it more attractive:
- The representative HE of the tumor and the margin of excision should be disclosed.
A1. We added the histological image and the postoperative MRI image in Figure 5 accordingly.
- The methods (assay, panel) used for NGS test should be mentioned in more detail. And the findings of NGS should be summarized.
A2. We added the sentences of explanation for the methods, and also added the figure of the result of copy number analyses (Figure 5C).
- The NGS test should include FGFR fusion and NTRK fusion.
A3. Unfortunately, we usually analyzed only copy number by NGS for resected glioma tissues because it is very important for diagnoses of adult diffuse gliomas. However, as the reviewer commented, we should have included such data molecular data. That will be more attractive. Thank you very much for your comments.
We have sought to the best of our ability to meet the high standards of Brain Sciences, and we believe that the constructive comments from the reviewer have improved the quality of our manuscript. However, we would be happy to revise the manuscript again, if necessary. Thank you very much for your time and efforts in reviewing our manuscript.
Sincerely,
Shunsuke Nakae, MD, MSc, PhD
Reviewer 2 Report
Thank you for involving me to review this manuscript. It's an excellent manuscript with a novel method of EEG monitoring on brain tumor-related epilepsy. However, I have some comments:
-
Anti-epileptic Drug: the terminology of AED has been changed to anti-seizure medication (ASM)
-
Line 54: What kind of seizure did she have? Seizures without loss of consciousness more than 10 times. Did the author mean 10 times/day? This kind of seizure should be written as “focal onset aware seizures” based on the 2017 seizure classification.
-
Line 55: “shaking her face left and right and slightly rotating her face to the right side..” Did the author means “focal face clonic movement (unilateral or bilateral?) and head turning to the right side”?
-
Line 65: “the administration of AED weas …” —> “the administration of AED was..”
-
Figure 1: should be self-explained by writing in the notes; i.e. Figure 1A: what was the sequence of the MRI? how many Tesla? what did MRI show? ; Figure 1B: What did CT scan show?
-
Line 81: “2.5% sevoflurane was…”. Please avoid writing numeric at the beginning of the sentence.
-
Line 82: “The reference electrode was placed on her forehead …”. Did author mean Fpz (based on 10-20 international system)? How many electrodes did author use? It should be written at the manuscript instead of the notes of figure 2.
-
Line 163: hypermotor seizure is different from focal onset aware seizure in this patient. Hypermotor seizure is primarily characterized by complex behavior involving proximal segments of the limbs and trunk, producing pedaling, kicking, pelvic thrusting, or rocking movements. Based on the semiology of the seizures in this patient, it is assumed that the patient had focal onset aware seizures which involving face muscles.
- Intractable epilepsy: this terminology has been changed to "drug-resistant epilepsy".
Author Response
The reviewers’ comments have helped us improve our manuscript. We have attempted to address all questions raised by the reviewer.
Thank you for involving me to review this manuscript. It's an excellent manuscript with a novel method of EEG monitoring on brain tumor-related epilepsy. However, I have some comments:
Anti-epileptic Drug: the terminology of AED has been changed to anti-seizure medication (ASM)
- It was corrected accordingly.
Line 54: What kind of seizure did she have? Seizures without loss of consciousness more than 10 times. Did the author mean 10 times/day? This kind of seizure should be written as “focal onset aware seizures” based on the 2017 seizure classification.
- I mean that the patient experienced the seizures more than ten times during the medical interview. I am sorry for confusing expression. I modified the sentences.
The reviewer’s comment is correct. I changed to the focal onset aware seizures accordingly.
Line 55: “shaking her face left and right and slightly rotating her face to the right side..” Did the author means “focal face clonic movement (unilateral or bilateral?) and head turning to the right side”?
- It was corrected accordingly.
Line 65: “the administration of AED weas …” —> “the administration of AED was..”
- Thank you very much for detecting the spelling error. It was corrected.
Figure 1: should be self-explained by writing in the notes; i.e. Figure 1A: what was the sequence of the MRI? how many Tesla? what did MRI show? ; Figure 1B: What did CT scan show?
- We added information about MRI and CT images accordingly in Figure 1. In addition, the CT image used in Figure 1B was changed so that the thickness of frontal bone was more easy to understand.
Line 81: “2.5% sevoflurane was…”. Please avoid writing numeric at the beginning of the sentence.
- It was corrected accordingly.
Line 82: “The reference electrode was placed on her forehead …”. Did author mean Fpz (based on 10-20 international system)? How many electrodes did author use? It should be written at the manuscript instead of the notes of figure 2.
- We do not mean FpZ.
We have already explained in our original manuscript how many electrodes were used in this study. Please see the following sentences from line 85 to 87: “Then, using the navigation-guided method, three deep electrodes with six electrodes each (Unique Medical Co., Ltd., Tokyo), were inserted into the tumor’s anterior, posterior, and lateral margins (Figure 2)”.
Line 163: hypermotor seizure is different from focal onset aware seizure in this patient. Hypermotor seizure is primarily characterized by complex behavior involving proximal segments of the limbs and trunk, producing pedaling, kicking, pelvic thrusting, or rocking movements. Based on the semiology of the seizures in this patient, it is assumed that the patient had focal onset aware seizures which involving face muscles.
- Thank you very much for your comments. We agree with reviewer’s comments, the manuscript was modified accordingly.
Intractable epilepsy: this terminology has been changed to "drug-resistant epilepsy".
- It was corrected accordingly.
We have sought to the best of our ability to meet the high standards of Brain Sciences, and we believe that the constructive comments from the reviewer have improved the quality of our manuscript. However, we would be happy to revise the manuscript again, if necessary. Thank you very much for your time and efforts in reviewing our manuscript.
Sincerely,
Shunsuke Nakae, MD, MSc, PhD
Reviewer 3 Report
The comments after reviewing the case report entitled “Applied fence-post techniques using deep electrodes instead of catheters for resection of glioma complicated with frequent epileptic seizures: A case report” has been provided as under, point by point.
Comments to authors.
The authors have summarized their study using combined navigation-guided fence-post techniques with continuous EEG monitoring during surgery for a patient with low-grade diffuse glioma complicated by frequent seizures, using deep electrodes as fence-posts instead of catheter tubes for epileptic seizures diagnosis and estimation of epileptogenic areas.
Apart from the minor mistakes in sentence structure, Abstract, keywords are correct however, the abstract should be re-written by avoiding the terms “I” or “we”. The same should be corrected throughout the MS.
Introduction and case presentation is OK, sufficient and up to date.
The tumor area should be highlighted/labeled in the figure 1 (MRI) for the viewers. The discussion is well organized and supported by relevant literature however, need some more updated. Over all, the MS seems to be concise.
Author Response
The reviewers’ comments have helped us improve our manuscript. We have attempted to address all questions raised by the reviewer.
The comments after reviewing the case report entitled “Applied fence-post techniques using deep electrodes instead of catheters for resection of glioma complicated with frequent epileptic seizures: A case report” has been provided as under, point by point.
Comments to authors.
The authors have summarized their study using combined navigation-guided fence-post techniques with continuous EEG monitoring during surgery for a patient with low-grade diffuse glioma complicated by frequent seizures, using deep electrodes as fence-posts instead of catheter tubes for epileptic seizures diagnosis and estimation of epileptogenic areas.
Apart from the minor mistakes in sentence structure, Abstract, keywords are correct however, the abstract should be re-written by avoiding the terms “I” or “we”. The same should be corrected throughout the MS.
- They were corrected as much as possible.
Introduction and case presentation is OK, sufficient and up to date.
The tumor area should be highlighted/labeled in the figure 1 (MRI) for the viewers. The discussion is well organized and supported by relevant literature however, need some more updated. Over all, the MS seems to be concise.
- We added the arrow, which indicates tumor location, in Figure 1A. We added more explanations in the manuscript.
We have sought to the best of our ability to meet the high standards of Brain Sciences, and we believe that the constructive comments from the reviewer have improved the quality of our manuscript. However, we would be happy to revise the manuscript again, if necessary. Thank you very much for your time and efforts in reviewing our manuscript.
Sincerely,
Shunsuke Nakae, MD, MSc, PhD
Round 2
Reviewer 1 Report
The revision is acceptable in the present form.